# Unsupervised Depth Estimation,
# 3D Face Rotation and Replacement

**Joel Ruben Antony Moniz**[1*], **Christopher Beckham**[2,3*], **Simon Rajotte**[2,3],
**Sina Honari**[2], **Christopher Pal**[2,3,4]
[1]Carnegie Mellon University, [2]Mila-University of Montreal, [3]Polytechnique Montreal, [4]Element AI
[1]jrmoniz@andrew.cmu.edu, [2]honaris@iro.umontreal.ca, [3]firstname.lastname@polymtl.ca

## Abstract

We present an unsupervised approach for learning to estimate three dimensional
(3D) facial structure from a single image while also predicting 3D viewpoint
transformations that match a desired pose and facial geometry. We achieve this
by inferring the depth of facial keypoints of an input image in an unsupervised
manner, without using any form of ground-truth depth information. We show how
it is possible to use these depths as intermediate computations within a new back-
propable loss to predict the parameters of a 3D affine transformation matrix that
maps inferred 3D keypoints of an input face to the corresponding 2D keypoints on
a desired target facial geometry or pose. Our resulting approach, called DepthNets,
can therefore be used to infer plausible 3D transformations from one face pose
to another, allowing faces to be frontalized, transformed into 3D models or even
warped to another pose and facial geometry. Lastly, we identify certain shortcom-
ings with our formulation, and explore adversarial image translation techniques as
a post-processing step to re-synthesize complete head shots for faces re-targeted to
different poses or identities. [1]

## 1   Introduction

Face rotation is an important task in computer vision. It has been used to frontalize faces for
verification [8; 19; 25; 28] or to generate faces of arbitrary poses [22; 18]. In this paper we present
a novel unsupervised learning technique for face rotation and warping from a 2D *source* image –
whose facial appearance will be used in the rotation – to a *target* face – to which the facial pose
and geometry inferred from the source image is mapped. A use case is when we have an image of
someone in a particular target pose and we want to put a given source face into that pose, without
knowing the exact target face pose. This can be leveraged, for example, in the advertisement industry,
when putting someone in a particular location can be costly or unfeasible, or in the movie industry
when the main actor's limited time or high cost can enforce using another actor whose face can be
later replaced by the main actor's. This is achieved through estimating the source face depth and
the 3D affine parameters that warp the source to the target face using neural networks. These neural
networks use a novel loss formulation for the structured prediction of keypoint depths. Once the 3D
affine transformation matrix is estimated, it can be used to warp the source image onto the target
face geometry using a textured triangular mesh. The use of a 3D affine transform means that we
can capture both a 3D rotation of the face to a new viewpoint as well as a global non-Euclidean
warping of the geometry to match a target face. We call these neural networks Depth Estimation-Pose
Transformation Hybrid Networks, or DepthNets in short.

*Our first contribution* is to propose a neural architecture that predicts both the depth of source
keypoints as well as the parameters of a 3D geometric affine transformation which constitute the

---

[*]Indicates equal contribution.
[1]Code will be released at: https://github.com/joelmoniz/DepthNets/

explicit outputs of the DepthNet model. The predicted depth and affine transformation could be then used to map a source face to a target face for object orientation, distortion and viewpoint changes.

*Our second contribution* consists of making the observation that given 3D source and 2D target keypoints, closed form least squares solutions exist for estimating geometric affine transformation models between these sets of keypoint correspondences, and we can therefore develop a model that captures the dependency between depth and the affine transformation parameters. More specifically, we express the affine transformation as a function of the pseudoinverse transformation of 2D keypoints in a source image – augmented by inferred depths – and the target keypoints. Thus, the second and major contribution in this work is capturing the relationship between an estimated affine transformation and the inferred depth as a deterministic relationship. In this formulation, DepthNet only predicts depth values explicitly and the affine parameters are inferred through a pseudoinverse transformation of source and target keypoints. Here, one can directly optimize through the solutions of what might otherwise be formulated as a secondary minimization step.

Our proposed DepthNet can map the central region of the source face to the target geometry. This leads to background mismatch when warping one face to another. Finally, *our third contribution* is to use an adversarial unpaired image-to-image transformation approach to repair the appearance of 3D models inferred from DepthNet. Together these contributions allow 3D models of faces that construct realistic images in the target pose. Our proposed method can be used for pose normalization or face swaps with no manually specified 3D face model. To the best of our knowledge, this is the first such neural network based model that estimates a 3D affine transformation model for face rotation which neither requires ground-truth 3D images nor any ground truth 3D face information such as depth.

## 2  Our Approach

As we have outlined above, our approach uses neural networks for inferring depth and geometric transformation – referred to as DepthNets; and, an adversarial image-to-image transformation network which improves the quality of the appearance of a 3D model inferred from a DepthNet.

### DepthNets

We propose three DepthNet formulations, described in Sections 2.1, 2.2, and 2.3. For each of the three models we explore two architectural scenarios: (A) a Siamese-like architecture that uses the source and target images themselves as well as keypoints extracted from these images, and (B) a fully-connected neural network variant which uses only facial keypoints in the source and target images. See Figure 1 (left) for details.

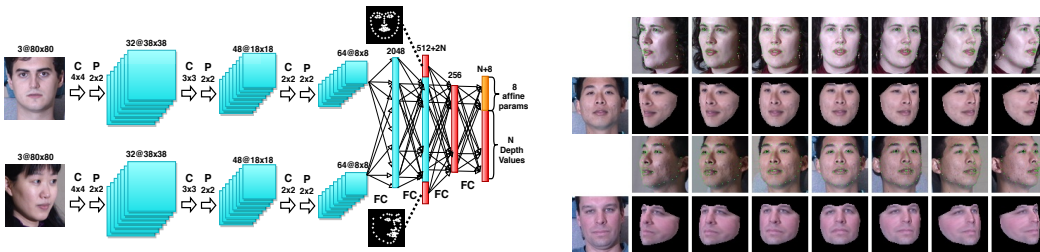

Figure 1: (Left) DepthNet architecture. The blue region is only used in case (A) and the red part is used in both cases (A) and (B), described in Section 2. The orange output (the 8 affine transformation parameters) is predicted only by model variations described in Sections 2.1 and 2.2, and not the model described in section 2.3. All three models predict the N depth values of the source keypoints. C, P, and FC correspond to valid conv, pool and fully-connected layers. The two paths of Siamese network share parameters and the black dots indicate concatenating keypoint values to FC units. (Right) Visualizing face rotation by re-projecting a frontal face (far left) to a range of other poses defined by the faces in the row above (in each pair of rows). In this experiment, we only use keypoints from the top-row in the DepthNet model (Model 7 in Table 1).

It is interesting to note that if DepthNets are used to register a set of images of objects to the same common viewpoint, the same image and geometry can be used as the target. This is the case for the frontalization of faces, for example. While the DepthNet framework is sufficiently general to be applied to any object type where 2D keypoint detections have been made, our experiments here focus on faces. We describe the three variants of DepthNets below.

## 2.1 Predicting Depth and Viewpoint Separately

In this variant of DepthNets, the model predicts both depths and viewpoint geometry, but as separate *explicit* outputs of a neural network. The input is comprised of only the geometry and pose of the source and target faces (encoded in the form of a 2D keypoint template), in case (B), or both keypoints and images of the source and target faces, in case (A). The key phases of this stage are described by the sequence of steps given below:

1. *Keypoint extraction*: Raw $(x, y)$ pixel coordinates corresponding to the keypoints of each image are extracted using a Recombinator Network (RCN) [10] architecture, and then concatenated before being passed into the keypoint processing step.

2. *(Optional) Image Feature Extraction*: DepthNets can be conditioned on only keypoints, case (B), or on keypoints and the original images, case (A). We can therefore optionally subject the source and target images to alternating conv-maxpool layers. If this component of the architecture is used, the last spatial feature maps in the Siamese architecture are concatenated before being given to a set of densely connected hidden layers.

3. *Keypoint processing*: In this step keypoints are passed through a set of hidden layers. If the *Image Feature Extraction* stage is used, the keypoints are concatenated to image features, the output of which is in turn fed to densely connected layers. The output layer of this phase will be of size $N + 8$, where $N$ is the number of keypoints. The first $N$ points represent the Depth proxy, and the last 8 points form a $4 \times 2$ matrix representing the learned parameters of the affine transform. See Figure 1.

4. *Geometric Affine Transformation Normalizer*: This phase applies the predicted affine transform on each (depth augmented) source keypoint to estimate its target location. Let $(x_s^i, y_s^i)$ represent the $i^{\text{th}}$ source keypoint, $(x_n^i, y_n^i)$ the corresponding source normalized keypoint estimated by applying the affine transformation matrix, $(x_t^i, y_t^i)$ the $i^{\text{th}}$ target keypoint (as ground truth (GT)), and $I_s$ and $I_t$ represent the source and target images respectively. Depending on which underlying architectural variant we use, two cases arise: one that utilizes only the keypoints (B), and another utilizing both the keypoints and the images (A). Since the keypoints are generated using RCNs, they are technically functions of the input images: $[x_s, y_s] = \mathbf{R}(I_s)$, and $[x_t, y_t] = \mathbf{R}(I_t)$. Depending on the (A) or (B) variant, the $i^{\text{th}}$ keypoint's predicted depth proxy $z_p^i$ is inferred as a function of the input keypoints, or both input keypoints and input images. In both cases the keypoints are derived from the images, so $\mathbf{z_p^i} = \mathbf{z_p^i}(I_s, I_t)$. Similarly, the 3D-2D affine transform $\mathbf{F}$ is a function of the images, such that $\mathbf{F} = \mathbf{F}(I_s, I_t)$, where the 8 predicted parameters are: $\mathbf{m} = \{m_1, m_2, m_3, t_x, m_4, m_5, m_6, t_y\}$. These constitute the 3D-2D affine transform which is used by all keypoints. In other words, each of the $i$ points is transformed using $x_n^i = \mathbf{F}(I_s, I_t)\, x_s^i$, or:

$$\begin{bmatrix} x_n^i \\ y_n^i \end{bmatrix} = \begin{bmatrix} m_1 & m_2 & m_3 & t_x \\ m_4 & m_5 & m_6 & t_y \end{bmatrix} \begin{bmatrix} x_s^i \\ y_s^i \\ z_p^i(I_s, I_t) \\ 1 \end{bmatrix}$$

The loss function of a DepthNet is obtained by transforming the source face to match the target face using the simple squared error of the corresponding target object's keypoint vector $\mathbf{x}_t = [x_t, y_t]^T$, as GT values, and the source object's normalized keypoint vector $[x_n, y_n]^T$. The loss for one example where we predict depth and affine viewpoint geometry can therefore be expressed as:

$$\mathcal{L} = \sum_{i=1}^{K} \left\| \mathbf{x}_t^i - \mathbf{F}(I_s, I_t)\, [x_s^i \ y_s^i \ \mathbf{z_p^i}(I_s, I_t)]^T \right\|^2 \tag{1}$$

5. *Image Warper*: This phase consists of using the depth proxy and affine transform matrix generated to actually warp the face from its source pose to be matched to the target object geometry. The final projection to 2D is achieved by simply dropping the transformed $z$ coordinate (which corresponds to an orthographic projection model). In the case of DepthNets, this orthographic projection is effectively embedded in the *Geometric Affine Transformation Normalizer* step, since the affine corresponding to the $z$ coordinate is not predicted, essentially dropping it.

As we operate on keypoints, the actual warping of pixels can be performed with a high quality OpenGL pipeline that performs the warp separately from the rest of the architecture. Source image, keypoints augmented with depth, and the affine matrix are passed to OpenGL pipeline to warp the source image towards the target pose. This OpenGL warping is not needed during DepthNet training, which means we do not have to do feedforward or backprop through OpenGL. In Summary, for step 1 the RCN model [10] is used, for steps 2 to 4 the DepthNet model, shown in Figure 1 (left), is trained,

and for step 5 an OpenGL pipeline is used. No data or parameters are needed to train the OpenGL pipeline. It warps images by directly using the provided data.

## 2.2 Estimating Viewpoint Geometry as a Second Step

In this model variant, training is similar to Section 2.1 and the model outputs depth and 3D affine transformation parameters. However, at test time, rather than using the predicted 3D affine transformation for pairs of faces, we use only the predicted depths and estimate the affine geometry parameters as a second estimation step. More precisely, given 3D points for a scene and the corresponding 2D points for a target geometry it is possible to formulate the estimation of a 3D affine transformation as a linear least squares estimation problem. An overdetermined system of the form $Am = x_t$ for this problem can be constructed as shown in (2).

This corresponds to an affine camera model followed by an orthographic projection to 2D keypoints. This setup also leads to the following closed form solution for the affine transformation parameters:

$$m = [A^T A]^{-1} A^T x_t, \qquad (3)$$

$$
\begin{bmatrix}
x_s^1 & y_s^1 & z_s^1 & 0 & 0 & 0 & 1 & 0 \\
0 & 0 & 0 & x_s^1 & y_s^1 & z_s^1 & 0 & 1 \\
x_s^2 & y_s^2 & z_s^2 & 0 & 0 & 0 & 1 & 0 \\
0 & 0 & 0 & x_s^2 & y_s^2 & z_s^2 & 0 & 1 \\
& & & \vdots & & & \\
x_s^K & y_s^K & z_s^K & 0 & 0 & 0 & 1 & 0 \\
0 & 0 & 0 & x_s^K & y_s^K & z_s^K & 0 & 1
\end{bmatrix}
\begin{bmatrix}
m_1 \\ m_2 \\ m_3 \\ m_4 \\ m_5 \\ m_6 \\ t_x \\ t_y
\end{bmatrix}
=
\begin{bmatrix}
x_t^1 \\ y_t^1 \\ x_t^2 \\ y_t^2 \\ \vdots \\ x_t^K \\ y_t^K
\end{bmatrix}
\qquad (2)
$$

where this pseudoinverse based transformation is parameterized by the reference points and their predicted depths.

## 2.3 Joint Viewpoint and Depth Prediction

Our *key observation* is that one can alternatively use the closed form analytical solution, measured in Eq. (3), for the least squares estimation problem as the underlying affine transformation matrix within the loss function. This leads to a special form of structured prediction problem for geometrically consistent depths and affine transformation matrix. For each image we have $\mathcal{L} =$

$$
\sum_{i=1}^{K} \left\| \underbrace{\begin{bmatrix} x_t^i \\ y_t^i \end{bmatrix}}_{x_t^i} - \underbrace{\begin{bmatrix} m_1 & m_2 & m_3 & t_x \\ m_4 & m_5 & m_6 & t_y \end{bmatrix}}_{m} \underbrace{\begin{bmatrix} x_s^i \\ y_s^i \\ z_p^i(I_s, I_t) \\ 1 \end{bmatrix}}_{x_s^i} \right\|^2 = \sum_{i=1}^{K} \left\| x_t^i - \text{reshape}\big[[A^T A]^{-1} A^T x_t\big] x_s^i \right\|^2
$$

where the matrix $A$ is parameterized as a function of $x_s$ as shown in Eq. (2). In this variant, the model *explicitly* outputs only depth values during train and test time. The affine transformation matrix in the equation above is replaced by Eq. (3), which measures the affine transformation as a pure function of source and target keypoints plus the inferred depth. The big difference of this formulation compared to Sections 2.1 and 2.2 is that geometric affine transformation parameters are no longer predicted by DepthNet during training and at both train and test time – it *solves* the least square loss through the pseudoinverse based transformation. Since $z_s^i = \mathbf{z_p^i}(\{x_s^j, y_s^j, x_t^j, y_t^j\}_{j=1...N})$ is predicted within the analytical formulation of the solution to the least squares minimization problem, we can backpropagate through the *solution* of a minimization problem that depends on the predicted depths. While we leverage keypoints for depth estimation, the proposed approach is novel in how the depth is estimated. Note that it is unsupervised with respect to depth labels. No depth supervision either by using depth targets (as in [2; 13; 15]), or by using depth in an adversarial setting (as in [24]), is used to estimate depth values for the base DepthNet models described in Sections 2.1, 2.2, and 2.3.

The depths learned for keypoints by these approaches are not necessarily true depths, but are likely to strongly correlate with the actual depth of each keypoint. This is because even though the method succeeds (as we shall see below) in aligning poses, the inferred depth and the affine transform may each be scaled by factors so as to cancel each other out (i.e., by factors which are multiplicative inverses of each other). Real world viewpoint geometry also involves perspective projection.

### Adversarial Image-to-Image Transformation

DepthNet transforms the central region of the source face to the target pose. Inevitably, the face background will be missing, which might make the proposed method unsuitable for many application

where the full face is required. To address this issue, we utilize CycleGAN [30], an adversarial image-to-image translation technique. This serves to repair the background of faces that have undergone frontalization or face swap through the DepthNet pipeline. Importantly, the adversarial nature of CycleGAN allows one to perform image transformation between two domains without the requirement of paired data. In our work, we perform experiments translating between various domains of interest but one example is translating between the domain of images in the dataset (i.e. the ground truth) and the domain of images where the DepthNet output is pasted onto the face region (in the case of face-swap). By doing so we clean the face background in an unsupervised manner.

# 3 Experiments

## 3.1 DepthNet Evaluation on Paired Faces

For the experiments in this section, we use a subset of the VGG dataset [16], with training and validating on all possible pairs of images belonging to the same identity for 2401 identities. This yields 322,227 train and 43,940 validation pairs. Check experimental setup details in Supplementary.

| Model | Color | MSE | MSE_norm |
|---|---|---|---|
| 1) A simple 2D affine registration | grey | 1.562 | 9.547 |
| 2) A 3D affine registration model using an average 3D face template | purple | 0.724 | 7.486 |
| 3) A DepthNet that separately estimates depth and geometry | brown | 0.568 | 6.292 |
| 4) The model above, but with a Siamese CNN image model | violet | 0.539 | 6.115 |
| 5) Secondary least squares estimation for visual geometry using the depths from 3) | red | 0.400 | 5.184 |
| 6) Secondary least squares estimation for visual geometry using the depths from 4) | green | 0.399 | 5.175 |
| 7) Backpropagation through the pseudoinverse based solution for visual geometry | orange | 0.357 | 4.932 |
| 8) The model above, but with a Siamese CNN image model | blue | **0.349** | **4.891** |

Table 1: (left) Comparing the Mean Squared Error (MSE) and MSE normalized by inter-ocular distance (MSE_norm) of different models. (right) Histogram of Mean Squared Errors. The second column in the Table (on left) corresponds to the color of the model in the figure (on right).

We explore the three variants of DepthNets described in Sections 2.1, 2.2, and 2.3, each with two architectural cases (A) and (B), depending on whether image features are used in addition to keypoints or not. We also compare with a number of baselines. We measure the mean square error (MSE) between the estimated keypoints on the target face (source face normalized keypoints) and ground truth target keypoints. Results for the following models are shown in Table 1:

1) A baseline model registrations using a simple 2D affine transformation.

2) We generate a 3D average face template from the 3DFAW dataset [14; 26; 6] by aligning the 3D keypoints of all faces in the dataset to a front-facing face using Procrustes superimposition. We report error by mapping the template face to each source face via Procrustes superimposition (to get a 3D face $f$) and then use an affine transformation from the 3D face $f$ to the target face.

3, 4) We use our proposed approach to predict both depth and geometry (described in Sections 2.1).

5, 6) These models described in Section 2.2. Note that during training, these two cases are similar to models 3 and 4 in Table 1.

7, 8) The pseudo-inverse formulation model described in Section 2.3.

As observed in Table 1, a simple 2D affine transform (model 1) without estimating depth and a template 3D face (model 2) get high errors on mapping to the target faces. DepthNet models get lower errors and the pseudo-inverse formulation (models 7 and 8) further reduces the error by 10%. The CNN models slightly reduce errors compared to their equivalent models that rely only on keypoints.

## 3.2 DepthNet Evaluation on Unpaired Faces and Comparison to other Models

In this section we train DepthNet on unpaired faces belonging to different identities and compare with other models that estimate depth. We use the 3DFAW dataset [14; 26; 6] that contains 66 3D keypoints to facilitate comparing with ground truth (GT) depth. It provides 13,671 train and 4,500 valid images. We extract from the valid set, 75 frontal, left and right looking faces yielding a total of 225 test images, which provides a total of 50,400 source and target pairs. We train the psuedoinverse DepthNet model that relies on only keypoints (model 7 in Table 1). We also train a variant of DepthNet that applies an adversarial loss on the depth values (DepthNet+GAN). This model uses a conditional discriminator that is conditioned on 2D keypoints and discriminates GT from estimated depth values. The model is trained with both keypoint and adversarial losses.

| Source \ Target | DepthNet | | | | DepthNet + GAN | | | |
|---|---|---|---|---|---|---|---|---|
| | Left | Front | Right | Avg | Left | Front | Right | Avg |
| Left | 24.67 | 27.71 | 29.70 | 27.36 | 59.78 | 59.67 | 59.63 | **59.69** |
| Front | 25.54 | 27.22 | 26.19 | 26.32 | 58.77 | 58.67 | 58.61 | **58.68** |
| Right | 21.66 | 21.48 | 23.87 | 22.34 | 59.97 | 59.70 | 59.60 | **59.76** |

Table 2: Comparing DepthCorr for different DepthNet models when mapping variant source to target poses. The Avg column measures the average over the three preceding columns.

We measure the correlation matrix between GT and estimated depths, where the element $k$ in the diagonal indicates the correlation between estimated and ground truth depth values for keypoint $k$, yielding a value between -1 and 1. We report the sum of absolute values of the diagonal of this matrix, indicated by DepthCorr. We compare DepthNet models on DepthCorr in Table 2. For this experiment we take every possible pair of source to target faces, where source and target are one of {left, front, right} looking faces. This yields a total of 5,550 pairs when the source and the target are from the same subset, and 5,625 pairs otherwise. This experiment measures the accuracy of depth estimation of the DepthNet models on different orientations of source-target faces. The baseline DepthNet model that does not leverage the depth labels performs well in different cases. DepthCorr improves more than twice for the DepthNet+GAN model, indicating a direct supervision loss using depth labels can enhance the depth estimation.

| Model | Need Depth | Manual Init. | MSE ($\times 10^{-5}$) | Depth Correlation Matrix Trace (DepthCorr) | | |
|---|---|---|---|---|---|---|
| | | | | Left pose | Front pose | Right pose |
| GT Depth | Yes | - | $8.86 \pm 6.55$ | **66** | **66** | **66** |
| AIGN [24] | Yes | No | $9.06 \pm 6.61$ | 44.08 | 50.81 | 49.04 |
| MOFA [20] | No | Yes | $8.75 \pm 6.33$ | 11.14 | 15.97 | 17.54 |
| DepthNet (Ours) | No | No | $\mathbf{7.65 \pm 6.97}$ | 27.36 | 26.32 | 22.34 |
| DepthNet + GAN (Ours) | Yes | No | $8.74 \pm 6.24$ | 59.69 | 58.68 | 59.76 |

Table 3: Comparing MSE and DepthCorr for different models. A lower MSE indicates the model maps better to the target faces. A higher DepthCorr indicates more correlation between estimated and GT depths.

We compare our two DepthNet models with three baselines: 1) AIGN [24], 2) MOFA [20] and 3) GT Depth (no model trained). AIGN estimates 3D keypoints conditioned on 2D heatmaps of the keypoints. MOFA estimates a 3D mesh using only an image. We implemented the AIGN model and asked the authors of MOFA to run their model on our test-set. They provided MOFA's results for 134 images in the test set. In Table 3 we compare these three models with our DepthNet models on DepthCorr. We also compare them on MSE, which is measured between GT and estimated target keypoints. Since the three baselines estimate depth on a single image due to their different model formulation, we first measure $m$ using closed form solution in Eq. 3 and then apply $m$ to the estimated source keypoints to get the target keypoint estimations. We contrast the estimated values with the GT target keypoints. As shown in Table 3, GT depth has the highest DepthCorr (the maximum possible value). The depths estimated by DepthNet+GAN and AIGN have stronger correlation to GT depth compared to the baseline DepthNet and MOFA, while baseline DepthNet performs better than MOFA. On MSE the baseline DepthNet model gets smaller MSE when mapping to target faces, indicating it is better suited for this task.

In Figure 2 we plot heatmaps of the estimated depth of different models (on Y axis) and the GT depth (on X axis) aggregated over all 66 keypoints on all test data. As can be seen, the depth estimated by DepthNet+GAN and AIGN models form a 45 degree rotated ellipses showing a stronger linear correspondence with respect to the GT depth compared to the the baseline DepthNet and MOFA.

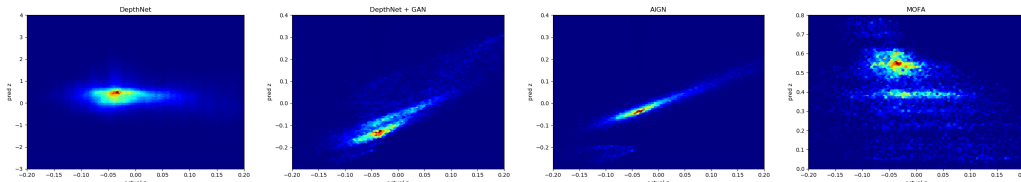

Figure 2: Predicted (Y axis) versus Ground Truth (X axis) depth heatmaps for different models.

In Figure 3 we show some estimated depth samples for different models (see more samples in Figure S1). AIGN and DepthNet+GAN generate more realistic results. MOFA generates very similar face templates for different poses. Baseline DepthNet estimates reliable depth values in most cases, however it has some failure modes as shown in the last row.

By comparing different models in Table 3, MOFA requires proper initialization to map face meshes to each image. AIGN requires depth labels to train the model. Our baseline DepthNet model neither require any depth labels nor any manual tuning. The results also show DepthNet can work well on unpaired data. We would also like to emphasize that MOFA and AIGN are designed to estimate a 3D model, while DepthNet is designed to estimate the parameters that facilitate warping a face pose to another without having depth values, so these models are designed to solve different problems.

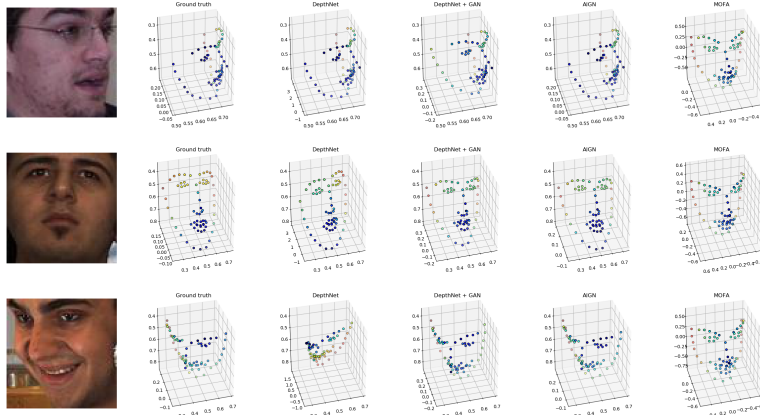

Figure 3: Depth visualization for different models (color coded by depth). From left to right: RGB image, Ground Truth, DepthNet, DepthNet+GAN, AIGN and MOFA estimated depth values.

An interesting observation is that GT depth gets a higher MSE compared to DepthNet. This can be due to not having a perspective projection between source and target faces. However, since DepthNet is trained to map to the target faces, it learns the affine parameters in a way to minimize this loss.

### 3.3 Face Rotation, Replacement and Adversarial Repair

In this section we show how DepthNet can be used for different applications. In Figure 1 (right) we visualize the face rotation by re-projecting a frontal face, from Multi-PIE [6], (far left) to a range of other poses defined by the faces in the row above. Since DepthNet (case B) computes transformation on keypoints rather than pixels it is robust to illuminations changes between source and target faces. See Figures S2 to S5 for further examples. Note that DepthNet preserves well the identity. However, it carries forward the emotion from source to target since using a global affine transformation imparts a degree of robustness to dramatic expression changes. The views in these figures are rendered from a 3D model in OpenGL. Note the model can align well to the target face poses.

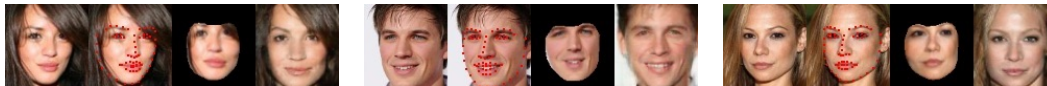

Figure 4: Background synthesis with CycleGAN. Left to right: source face; keypoints overlaid; DepthNet (DN); DN + background → frontal;

In another experiment, we do face frontalization with synthesized background. Here we use CycleGAN to add background detail to a face that has been frontalized with DepthNet. Referring to Figure 4, we perform this by conditioning the CycleGAN on the DepthNet image (column 3) and the background of column 2 (masking interior face region determined by the convex hull spanned by the keypoints). The second domain contains ground truth frontal faces. This experiment shows how to leverage DepthNet for full face generation. Note that we do not use identity information in this experiment. However, it can be used to better preserve the identity.

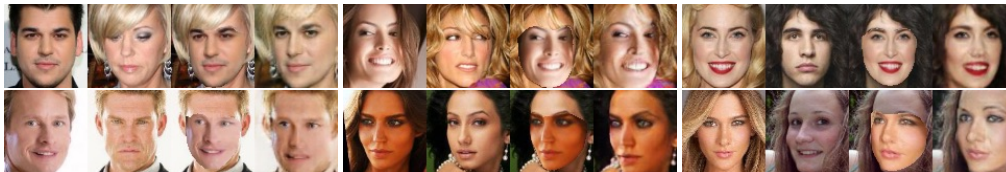

Figure 5: Left to right: source face; target face; warp to target; repaired result.

Finally, we do face swaps, where we warp the face of one identity onto the geometry and background of another identity using DepthNet. To do so, we paste the rotated face by DepthNet onto the background of the target image and train a CycleGAN to map from the domain of 'swapped in faces' to the ground truth faces in our dataset, effectively learning to clean up face swaps so that the face region matches the hair and background. Some examples of this procedure are shown in Figure 5.

# 4   Related Work

## 4.1   3D Transformation on Faces

While there is a large body of literature on 3D facial analysis, many standard techniques are not applicable to our setting here. As an example, morphable models [1] cover a wide variety of approaches which are capable of high quality 3D reconstructions, but such methods usually require 3D face scans or reconstructions from multi-view stereo to be assembled so as to learn complex parametric distributions over face shapes. A close approach to our own is that of [7] on viewing real world faces in 3D. Similar to our work, this approach does not require aligned 3D face scans, highly engineered models or manual interventions. They make the observation that if 2D keypoints can be obtained from a single input image of a face and these keypoints are matched to an arbitrary 3D target geometry, then standard camera calibration techniques can be used to estimate plausible intrinsics and extrinsics of the camera. This allows the estimated camera matrix, 3D rotation matrix and 3D translation vector to be used to transform the target 3D model to the pose of the query image from which an approximate depth can be obtained. Hassner et. al [8] explore the use of a single unmodified 3D surface as an approximation to the shape of all input faces. In contrast, our approach only requires 2D keypoints from the source and target faces as input. It then estimates the depth of the source face keypoints, thereby inferring an image specific 3D model of the face.

DeepFace [19] uses face frontalization to improve the performance of a face verification system. It uses a 3D mask composed of facial keypoints, detects the corresponding locations of these keypoints in the image, and maps the 2D keypoints onto a 3D face model to frontalize it. DeepFace, however, maps to a template 3D face, therefore always mapping to a specific pose and geometry. DepthNet, on the other hand, can map to any pose and geometry, giving it more expressive flexibility.

## 4.2   Generative Adversarial Networks on Face Rotation

Recently, adversarial models in [12; 22; 25; 27; 18; 28] have explored face rotation. TP-GAN [12] performs face frontalization through introducing several losses to preserve identity and symmetry of the frontalized faces. PIM [28] frontalizes faces in a composed adversarial loss and then extracts pose invariant features for face recognition. These models are mainly aimed for face verification, where they can only do face-frontalization. Another limitation of these models is in requiring ground truth frontal images of the same identity during training. DR-GAN [22] rotates faces to any target pose by using a discriminator that also does identity classification in addition to pose prediction, to preserve id and pose. While these models do pure face rotation of a 2D face, our model can warp the input face to any other target face, allowing warping the input face to any other identity, with a different geometry and pose. Moreover, our model also estimates the 3D geometric affine transformation parameters explicitly, allowing these parameters to be used later, e.g., for face texture swap.

FF-GAN [25], DA-GAN [27], and FaceID-GAN [18] estimate parameters of either a 3D Morphable Model (3DMM), as in [25; 18], or source to target pose transformation, as in [27]. FF-GAN uses 3DMM parameters to frontalize faces in an adversarial approach, while FaceID-GAN uses the 3DMM parameters to generate any target pose. These models, however, train 3DMM on ground truth labels such as identity, expression and pose. DepthNet, on the other hand, estimates depth and affine transformation parameters without requiring ground truth affine or depth labels or pre-training. Similar to DepthNet, DA-GAN [27] estimates parameters of an affine transformation model that maps a 2D face to a 3D face. Unlike DepthNet that estimates depth on the source face, DA-GAN uses depth in a template target face. While their approach eliminates the need for depth estimation, it only allows the source face to be mapped to the target template geometry, while DepthNet can map the source face to any target geometry, provided by a target image, or its keypoints. We demonstrate the application of this flexibility for the face replacement task.

The aforementioned adversarial models use an identity preserving loss to maintain identity. The core DepthNet model does not need identity labels and preserves well the identity (as shown in Figure 1

(right)). However, the identity information can be used by the proposed adversarial components, as in background synthesis, to further improve the results. Unlike some of these models that take target pose as input, DepthNet uses the target keypoints to estimate the target geometry and does not require the target pose. This has several advantages; 1) DepthNet can map to the geometry of the target face in addition to the pose, and 2) in the face replacement task, DepthNet can replace the target face with the warped source face directly onto the target face location. Its application is shown in the face swap experiment in Section 3.3.

### 4.3 Depth Estimation

Thewlis et. al [21] propose a mapping technique to learn a proxy of 2D landmarks in an unsupervised way. A semi-supervised technique has been also proposed in [11] that improves landmark localization by using weaker class labels (e.g. emotion or pose) and also by making the model predict equivariant variations of landmarks when such transformations are applied to the image. Similar to these approaches, DepthNet also maps a source to a target to learn its parameters. However, unlike these two approaches that estimate 2D landmarks, DepthNet estimates the depth of the landmarks using 2D matching of keypoints, by formulating affine parameters as a function of depth augmentated keypoints in a closed form solution.

While several models [13; 2; 15] estimate depth with direct supervision, there has been recent models [29; 5; 3] that estimate depth in an unsupervised training procedure. These models rely on pixel reconstruction by using frames that are captured from very similar scenes, e.g. nearby frames of a video [29] or left-right frames captures by stereo cameras [5; 3]. These models estimate depth on one frame and then by using the disparity map, measure how pixel values of nearby frames compare to each other. To do this, they also require camera intrinsic parameters, e.g. focal length or distance between cameras. Unlike these models, our approach does not require source to target pixel mapping. This allows mapping faces from different people with completely different skin colors, without knowing camera parameters or how they are positioned with respect to each other. Therefore, DepthNet is not susceptible to variations in illuminations or lighting between source and target faces.

Tung et. al [23] estimate 3D human pose in videos, where they use synthetic data to pre-train internal parameters of the model and fine-tune them by keypoint, segmentation and motion loss. Adversarial Inverse Graphics Networks (AIGN) [24] estimates 3D human pose from 2D keypoint heatmaps in a semi-supervised manner with a similar formulation to that of CycleGAN. They apply an adversarial loss on the 3D pose to make them look realistic. These models leverage the depth values either through synthetic data [23], or by adversarial usage of ground truth depth values [24]. Unlike these models, DepthNet does not rely on any depth signal, either directly or indirectly. MOFA [20] builds a 3D face mesh using a single image, where the 3D face parameters such as 3D shape and skin reflectance are estimated by an encoder and then using a differentiable model they are rendered back to the image by the decoder. This model requires manual initialization to map the input image to the 3D mesh, since otherwise it is doing an unconstrained optimization by adapting both the face pose and the skin reflectance. Our model, however, does not require any manual initialization.

## 5 Conclusion

We have proposed a novel approach to 3D face model creation which enables pose normalization without using any ground truth depth data. We achieve our best quantitative keypoint registration results using our novel formulation for predicting depth and 3D visual geometry simultaneously, learned by backpropagating through the analytic solution for the visual geometry estimation problem expressed as a function of predicted depths. We have illustrated the quality and utility of the depths and 3D transformations obtained using our method by transforming source faces to a wide variety of target poses and geometries. Our technique can be used for face rotation and replacement and when combined with adversarial repair it can blend warped faces to also synthesize the background. The proposed model, however, carries forward emotion from source to target due to learning a shared affine parameters for all keypoints. Moreover, for extreme non-frontal faces, while DepthNet can extract the transformation params (since it only relies on keypoints), OpenGL cannot extract texture due to occlusion. We show an example of how to address this in the supplementary material. An interesting extension to this paper can be replacing the OpenGL pipeline with a generative adversarial framework that synthesizes a face using the parameters estimated by DepthNet.

# 6  Acknowledgments

We would like to thank Samsung and Google for partially funding this project. We are also thankful to Compute Canada and Calcul Quebec for providing computational resources, and to Poonam Goyal for helpful discussions.

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
