[Supplementary Material]

# Unsupervised Depth Estimation, 3D Face Rotation and Replacement: Supplementary Information

## S.1 Experimental Setup

The RCN, the DepthNets and the CycleGAN modules are trained separately. Each model is trained using standard techniques for the model class and has a separate objective to be optimized. DepthNet does not use the OpenGL pipeline during training and only uses it to render faces at test time, allowing DepthNet to train faster.

### S.1.1 DepthNet Experimental Details

Here we describe the details of the experiments carried out in Section 3.1. Our DepthNet architectures require keypoints of both source and target images to be extracted. For this, the image is first passed through the VGG-Face [16] face detector. The face crops are then scaled down to $80 \times 80$ and converted to greyscale, following which they are passed through RCN to obtain $N = 68$ keypoints on each image. The RCN is trained exactly as described in [10], using the 300W dataset [17].

The keypoint only variant of our model involves concatenating all detected keypoints and passing them through a two-layer deep fully connected network, with 256 hidden units and $o$ output units. The size of $o$ depends on whether we are predicting only the depth, in which case $o = N$, or both depth and affine transformation parameters, in which case $o = N + 8$.

As discussed above, it is possible to augment these models with a Siamese CNN module (case A). In the model variants that also use the image, we pass both the source and the target images through three conv-maxpool layers with shared weights of size (32, 4, 2), (48, 3, 2), (64, 2, 2), respectively for the representation (`num_filters`, `filter_size`, `pool_size`). The network's outputs for the source and target faces are then concatenated before passing them into a 4-layered fully connected network with respective output sizes of 2048, 512, 256, and $o$. The keypoints are concatenated to the 512-unit layer before being passed to the last two layers. See Figure 1 (left) for an illustration of the model. We explore these Siamese CNN augmented variants in models 4, 6 and 8 in Table 1.

We set the initial learning rate to 0.001 and use a Nesterov momentum optimizer (with a momentum of 0.9) in all our experiments. With the exception of the last layer, we initialize all weights with a Glorot initialization scheme [4], with the weights sampled from a uniform distribution. We use a ReLU gain [9], set all biases to 0, and apply a ReLU non-linearity after every layer. In the final output layer, we do not apply any non-linearity and initialize the weights to 0. The biases of units that represent depths are initialized to a random Normal distribution with $\mu = 0$ and $\sigma = 0.5$, while those that form the predicted affine transform are initialized with the equivalent of a "flattened" identity transform. All models have been trained for 500 epochs.

We point out that except for a comparison between learning rates in the set {0.01, 0.001, 0.0001} over few (less than 10) epochs, to find a learning rate that the model seems to train well with, we have not performed a hyperparameter search, and anticipate that the performance of the model can be made *even* better by searching the hyperparameter space on a per model basis and by using deeper (or modified) architectures.

## S.2 Depth Visualization

We show further estimated depth values by different models in Figure S1. DepthNet+GAN and AIGN generate the closest depth values to the GT depth. The baseline DepthNet model estimates reliable depth values for most cases, however it has some degree of inaccuracy, as shown in the last two rows. In DepthNet, the estimated depth indicates the position of each keypoint relative to other keypoints rather than with respect to a source and importantly it is done without any supervision. MOFA, which is also unsupervised, generates very similar face templates for different cases.

Figure S1: Depth visualization for different models (color coded by depth). The Depth axis is the one pointing into the page. From left to right: RGB image, Ground Truth, DepthNet, DepthNet+GAN, AIGN and MOFA estimated depth values.

## S.3  Additional Camera Sweep Visualizations

In this section, we present additional visualizations along the lines of those shown in Figure 1 (right) in Section 2 of the main paper. Frontal faces selected from the Multi-PIE dataset [6] are re-projected to match several other poses corresponding to a person with a different identity. The DepthNets predicts reliable proxy depths, which when coupled with the analytically obtained affine transform (obtained from the least-squares pseudo-inverse-based solution described in Section 2.3 of the main paper) yields faces close to the desired target face geometry when passed through the OpenGL pipeline.

We show camera sweeps for frontal source faces in Figures S2 and S3 and non-frontal source faces in Figure S4. In Figure S5 we use the same target identity as the source face, showing how much the

Figure S2: Projecting a frontal face (far left) to a range of other poses defined by faces in the row above.

generated face differs from the ground truth target. For all samples in Figures S2, S3, S4, and S5 we use the DepthNet model that relies on only key-points (model 7 in Table 1).

Note that for non-frontal source faces, the quality of images is reduced specially for frontal target faces. This is due to lack of adequate texture on the occluded side of the face to be transferred to the target pose by OpenGL pipeline, rather than inaccuracies in the affine transformation parameters. In order to reduce the side-affects, we use either of the source face or its flipped version, that is closer to the target face pose, and then warp the face to the target keypoints.

Figure S3: Projecting a frontal face (far left) to a range of other poses defined by faces in the row above.

Figure S4: Re-projecting a non-frontal face (far left) to a range of other poses defined by faces in the row above.

Figure S5: Rotating a face (far left) to a range of other poses defined by faces of the same identity in the row above. On the top row frontal and on the bottom row non-frontal source faces are shown.

## S.4  CycleGAN

Suppose we have some images belonging to one of two sets $\mathbf{x} \in X$ and $\mathbf{y} \in Y$, where $\mathbf{x}$ denotes a DepthNet-resulting face and $\mathbf{y}$ a ground truth face which is frontal. We wish to learn two functions

$F : X \to Y$ and $G : Y \to X$ which are able to map an image from one set to the corresponding image in the other. Correspondingly, we have two discriminators $D_X$ and $D_Y$ which try to detect whether the image in that particular set is real or generated. While we are only interested in the function $F : X \to Y$ (since this is mapping to the distribution of ground truth faces) the formulation of CycleGAN requires that we learn mappings in both directions during training. We optimize the following objectives for the two generators $F$ and $G$:

$$\min_{G,F} \mathbb{E}_{\mathbf{x},\mathbf{y}} \Big[ \ell(D_X(G(\mathbf{y})), 1) + \ell(D_Y(F(\mathbf{x})), 1) + \lambda||\mathbf{y} - F(G(\mathbf{y}))||_1 + \lambda||\mathbf{x} - G(F(\mathbf{x}))||_1 \Big] \quad (4)$$

And the following for the two discriminators $D_X$ and $D_Y$:

$$\min_{D_X, D_Y} \mathbb{E}_{\mathbf{x},\mathbf{y}} \Big[ \ell(D_X(\mathbf{x}), 1) \; + \; \ell(D_X(G(\mathbf{y})), 0) \; + \; \ell(D_Y(\mathbf{y}), 1) \; + \; \ell(D_Y(F(\mathbf{x})), 0) \Big], \quad (5)$$

where 0/1 denote fake/real, $\ell()$ is the squared error loss and $\lambda$ is a coefficient for the cycle-consistency (reconstruction) loss.

In the case where we did adversarial background synthesis, $\mathbf{x}$ is a channel-wise concatenation of the DepthNet-frontalized face and the background of the original (pre-frontalized) image. For face-swap cleanup, $\mathbf{x}$ is simply a source face which has been warped to a target face and pasted on top.

Once the network has been trained, we can disregard all other functions and use $F$ to clean up faces which are low quality due to artifacts from warping.

In terms of architectural details the generators and discriminators used were those described in the appendix of the CycleGAN paper [30]. In short, the generator consists of three `conv-BN-relu` blocks which downsample the input, followed by nine ResNet blocks (which can be interpreted as iteratively performing transformations over the downsampled representation), followed by `deconv-BN-relu` blocks to upsample the representation back into the original input size. For training, we use the same hyperparameters as most CycleGAN implementations which is using the Adam optimizer with learning rate $\alpha = 2 \times 10^{-4}$, $\beta_1 = 0.5$, $\beta_2 = 0.999$. However, instead of using a batch size of 1 we use the largest possible batch size, which was 16 for a 12GB GPU.

Note that in order to produce better translations, the dataset we used for all CycleGAN experiments contain both the VGG and the CelebA datasets, which has significantly more images.

### S.4.1 Background Synthesis

We present extra visualizations for the CycleGAN which performs background synthesis on CelebA, corresponding to Figure 4 in the main paper. These are shown in Figure S6.

Figure S6: Background synthesis with CycleGAN. Left to right: source face; keypoints overlaid; DepthNet (DN); DN + background $\to$ frontal

### S.4.2 Face Replacement and Adversarial Repair

In Figure S7 we provide extra face swap samples on CelebA, corresponding to Figure 5 in the main paper, where a source face is warped to a target face pose using DepthNet, pasted onto the

target image and then passed to a CycleGAN to adapt the face skin of the warped source face to the background and hairstyle of the target face.

Figure S7: Face swap experiment with CycleGAN. Left to right: source face; target face; warp to target with DepthNet; repaired result with CycleGAN. The source face is taken and warped onto the target face. The background and hairstyle is then adapted to the target face.

### S.4.3 Extreme Pose Face Clean-up

If the source image has an extreme pose, the texture will be missing on the occluded side of the face and the OpenGL pipeline cannot rotate the face without artifacts. Note that this shortcoming is due to lack of texture on the occluded side of the face rather than a deficiency of the transformation parameters measured by DepthNet.

We performed an experiment using CycleGAN to fix such artifacts. For this experiment we take source images from CelebA and first frontalize it by using DepthNet. Since the frontalized faces have artifacts due to stretch of texture on the occluded side of the face by the OpenGL pipeline, we train a CycleGAN that takes DepthNet frontalizaed faces plus the background of the original non-frontal image (as two images) in one domain and the ground truth frontal faces in the other domain. The CycleGAN learns to clean-up these artifacts. Finally, we take the GAN-repaired frontalized faces and project it to different target poses using DepthNet. In Figure S8 we visualize camera sweep for source faces in the wild that have extreme poses. The cycleGAN reasonably cleans the face artifacts and then DepthNet projects it to different poses. This is just one approach to address the extreme pose occlusion artifacts. We see alternative methods for addressing this issue as promising directions for future research.

Figure S8: Re-projecting a non-frontal face (far left) from CelebA to a range of other poses defined by faces in the row above. Top row (in each pair) depicts the target faces from Multi-PIE [6]. The bottom row shows from left to right: source face, souce face frontalized by DepthNet, adversarial-repaired face, the repaired source face projected to the target poses (4th to 10th columns).