[Reviews · NeurIPS 2018]

Reviewer 1



Summary: This paper propose an unsupervised approach to estimate 3D facial structure from a single image while can predict 3D transformations that match a desired pose and facial geometry. A Siamese-like architecture is used to predict relative affine transform between the two given images and depth of keypoints of each one, based on which a matching loss is computed which drive the learning of the networks. Novelty: This work propose a novel explicit 3D structural understanding for deformable face through 2D image matching (with or without keypoints representation) by inducing affine transformation with a direct closed-formed solution. It could be seen as an extension to unsupervised rigid scene based works, and has potential to further extend to articulated ones like human pose etc. Reference: Overall, it is complete, and I think it could be related with Thewlis et.al. Unsupervised learning of object landmarks by factorized spatial embeddings. ICCV 2017, and their extensions. Both of the works use similar matching strategy, while the one I mentioned is more implicit. Experiments: In general, it is complete by providing different variations of the method and out perform previous state-of-the-art (MOFA) by a relative large margin. The only thing it needs some adjustment of table and figure since they are to small to look.

Reviewer 2



This paper presents a method to estimate 3D facial structure (the depth of facial keypoints) from a single image while also predicting 3D viewpoint transformations that match a desired pose and facial geometry without using any ground truth depth data (unsupervised learning). The authors show the possibility to use these depths as intermediate computations within a new backpropable loss to predict the parameters of a 3D affine transformation matrix that maps inferred 3D keypoints of an input face to the corresponding 2D keypoints on a desired target facial geometry or pose. The proposed method can be used to infer plausible 3D transformations from one face pose to another, allowing faces to be frontalized, transformed into 3D models or even warped to another pose and facial geometry. Thorough experimental results on VGG, 3DFAW and Multi-PIE demonstrate the effectiveness of the proposed method. I believe this paper shows a promising approach for the face-related applications (e.g., depth estimation, 3D face rotation, replacement, face frontalization, pose-invariant face recognition) that I have not seen elsewhere so far. The paper is written clearly, the math is well laid out and the English is fine. I think it makes a clear contribution to the field and can promote more innovative ideas, and should be accepted. Additional comments: - The paper is missing discussions and comparisons with some related works and also does not cite several related works, e.g., FF-GAN [Yin et al., ICCV 2017], DA-GAN [Zhao et al., NIPS 2017], FaceID-GAN [Shen et al., CVPR 2018], PIM [Zhao et al., CVPR 2018], etc. - It is not clear how the proposed method performs on instance of large variations in expressions, lighting and illumination. - How did authors update each component and ensure stable yet fast convergence while optimising the whole framework?

Reviewer 3



[Summary] This paper proposes a method to project a source facial image to a target facial image with the help of 2D keypoints (or, facial landmarks). In particular, the method estimates the depth of the 2D keypoints of the source images using information from both images, and the method estimates the 3D-to-2D affine transform from the source to the target. With this transformation, a traditional keypoint-based face warping (implemented in OpenGL) algorithm and CycleGAN are used to map the source image to the target image. Note that the estimation of the depth and affine transform can either depends on only the 2D keypoints or both the keypoints and images. [Pros] The paper proposes three slightly different ways to estimate the depth of 2D keypoints and the 3D-to-2D transformation from the source image to the target image. The formulation of all the variants are reasonable, and explicitly predicting depth as an intermediate step is interesting. Detailed comparisons are performed among these variants to justify the effectiveness of the final model, which estimates the depth first and incorporate the keypoint-based estimation of the affine transform into the neural network. The results in Figure 4 for replacing faces are interesting. [Cons] 1. It is unclear why the paper addresses an useful task. The problem setting assumes the keypoints or also the images for both the source and target faces are known (or, detected for the keypoints). a) Suppose both images and keypoints are given (case A in the paper), why projecting the first image to the second image is a meaningful task? b) When only the keypoints are given for the target face (case B), there are also many existing works for face morphing. Why estimating an affine transform first and then transform the source face is a better way of face morphing? c) For specific morphing tasks like frontalization, the minimal information needed is the rotation angle. Requiring keypoints for the target face are unrealistic or costly. If the keypoints are detected, we have already known the target images (refer to 1.a) … If the keypoints are given manually, it is too costly. (It is possible that I misunderstood the experimental setting for the frontalization, which was not well-clarified in the experimental section.) 2. The paper emphasized “unsupervised learning,” but the method does not seems to be “unsupervised.” a) The acquisition of keypoints (either a pretrained detector or ground truth) are obtained from supervision. So, at least for the face rotation and replacement, the setting is not fully unsupervised. b) The models are trained using paired facial images from the same identities. It is already a type of supervision to know if images are from the same identity or not. Most experimental results in this paper are about keypoints. The results for the tasks of face rotation are limited. And the qualitative results in Figure 3 are not visually strong. The identity does not seem to be preserved well during the rotation. Also, no experiments are present to show the usefulness of the frontalized images. Experiments on facial recognition may be helpful. There are existing works for both face frontalization and replacement, but no qualitative or quantitative comparison is performed. [Overall] The paper proposed interesting techniques, but it is not clear what are the central problems addressed by these techniques and what insights are provided. The experimental results focused mainly on the intermediate results (i.e., the keypoint mapping and keypoint-depth estimation). The results for face frontalization and replacement are limited. -------------------------- Thank you for the authors’ response. The response tried to address the major concerns I have. I agree with the author that the key-point transform can be useful for movie making, but I still feel the significance of the method is limited and not sufficiently argued. Essentially, the paper address the problem of a generalized stereo-based 3D reconstruction for key-points using a learning-based method. The estimated transform between the two sets of key-points is applied to face frontalization and face replacement. However, a large number of works exist on stereo-to-3D reconstruction for both sparse key-points and dense images. Stereo-based face reconstruction is particularly well-studied. What might be interesting is that the proposed method does not necessarily require the two images to be captured at the same time and potentially not for the same person. But given the consistent structure of human faces, the results are not that surprising. I acknowledge the authors' explanation about the “unsupervised,” which was also my understanding of this paper. However, the title and abstract could mislead people to believe this paper is addressing unsupervised 3-D key-point learning problem. Moreover, in this misunderstood case, the paper could be more relevant to Thewlis et al.’s ICCV paper (which was suggested by R1). The face frontalization and replacement are pretty separate from the depth and transform estimation. It can be also done with the depth and transform estimated by other stereo-to-3D methods. However, the paper lacks comparison with other stereo-to-3D methods using the same image generator. Moreover, it also lacks comparison with face frontalization methods in other frameworks. Overall, the paper does not define its problems clearly and shows a tendency to mix up things. So while appreciating the authors’ efforts in the rebuttal, I lean to reject this paper and hope the paper can get improved in the future.